The development of curvature in the porcine radioulna

Pantinople Jess
McCabe Kyle
Henderson Keith
Richards Hazel L. hazel.l.richards@gmail.com
Milne Nick
School of Anatomy, Physiology and Human Biology, University of Western Australia , Perth, WA , Australia
Abdala Virginia
Electronic publication date: 2017 Jun 1
Publication date: 2017
Volume: 5
Electronic Location ID: e3386
Received 2016 Dec 14; Accepted 2017 May 7
Copyright: © 2017 Pantinople et al.
Copyright year: 2017
Copyright holder: Pantinople et al.
License: This is an open access article distributed under the terms of the Creative Commons Attribution License, which permits unrestricted use, distribution, reproduction and adaptation in any medium and for any purpose provided that it is properly attributed. For attribution, the original author(s), title, publication source (PeerJ) and either DOI or URL of the article must be cited.
License URL: https://creativecommons.org/licenses/by/4.0/

Keywords: Development, Radioulna, Chondral modelling, Habitual loading, Bone curvature, Pig

Funding: The authors received no funding for this work.

==============================
Long bone curvature in animal limbs has long been a subject of interest and much work has explored why long bones should be curved. However, the ‘when’ and ‘how’ of curvature development is poorly understood. It has been shown that the rat tibia fails to attain its normal curvature if the action of muscles is removed early in life, but it is not clear if this is because the curvature fails to develop or if the bone becomes straighter without the action of muscles. No studies have examined the development of bone curvature in a normally developing quadruped, so this study tracks the course of curvature formation in the radioulna in a series of growing pigs. We also histologically examined the epiphyseal growth plates of these bones to determine if they contribute to the formation of curvature. In all three epiphyseal plates examined, the proliferative zone is thicker and more densely populated with chondrocytes on the cranial (convex) side than the caudal (concave) side. Frost’s chondral modelling theory would suggest that the cranial side of the bone is under more compression than the caudal side, and we conclude that this is due to the action of triceps extending the elbow by pulling on the olecranon process. These results support the idea that bone curvature is an adaptation to habitual loading, where longitudinal loads acting on the curved bone cause bending strains that counter the bending resulting from the habitual muscle action.

Introduction

Long bone curvature in animal limbs has been a subject of interest because it would seem to weaken the bone under weight-bearing conditions—that is, theoretically, curved bones are more prone to failure than straight bones (Bertram & Biewener, 1988). Many studies have used strain gauges in vivo to examine the strains experienced by curved bones during locomotion (Lanyon & Baggott, 1976; Lanyon & Bourn, 1979). More contemporary studies have used finite element analysis to model these strains under load (Jade et al., 2014; Milne, 2016), while others have studied the relationship between body size and bone curvature (Biewener, 1983; Swartz, 1990; Shackelford & Trinkaus, 2002). It is widely understood that altering the mechanical environment of bones affects their growth (Mosley, 2000) and metabolism (Uhthoff & Jaworski, 1978)—for instance, Lanyon (1980) showed that rat tibiae failed to attain normal curvature if loads were removed by sciatic neurectomy and patella tenotomy.

In their study, Lanyon (1980) stated that the tibia ‘fails to attain’ its normal curvature, but did not clarify whether the bone was initially straight and did not develop its normal curvature, or if it was initially curved and became straighter due to the intervention. Searching the literature failed to reveal a clear account of when and how the curvature of such bones appears. Therefore, the first aim of this study is to document when bone curvature develops, using radioulnae from a series of foetal through to six-month-old pigs (Sus scrofa domesticus). If the curvature of the radioulnae increases as the young pig grows and begins to use its forelimbs in locomotion, the idea of curvature development as a response to locomotor loading will be supported.

The second question addressed here is how such curvature develops. There are two ways a bone may alter its shape or size: endochondral growth during development and surface remodelling. Frost (1973) described how bones might adjust their shape by surface remodelling in response to loading. Indeed, a curved bone under longitudinal loading would be expected to become straighter, because the concave surface would be under compression—and thus depository—and the convex side would be under tension and thus resorptive. Such a bone would straighten by surface remodelling (drift) under these loads. Conversely, Frost described how an eccentrically loaded bone may become curved by the same mechanism (Frost, 1964).

Developing bones have epiphyseal growth plates which are the principal means of bone elongation. Frost (1979) described a chondral modelling theory—he suggested that articular or epiphyseal growth cartilages respond to differences in loading: growing faster under physiological compression and slower under tension. Asymmetric growth at the distal femoral epiphysis has been shown to contribute to the orientation of the knee, producing the carrying angle in humans (Tardieu & Trinkaus, 1994). Cartilage can also form at the site of repair in bone fractures, and it has been demonstrated, in newborn mice, that these chondrocytes respond in the same way to tension and compression as those concerned with epiphyseal growth (Rot et al., 2014). This is the mechanism by which these bones can straighten during fracture repair (Rot et al., 2014). It is possible that such uneven chondrocyte activity, in response to compressive and tensile loading, may be a way of producing curvature during development.

So, our second approach in this study is to examine the cranial and caudal sides of the epiphyseal plates of the porcine radius and ulna, and measure the thickness of the proliferative (growth) zone as well as the density of chondrocytes within this zone. This zone is identifiable as the area where chondrocytes appear flattened and are arranged in columns (Niehoff et al., 2004). If the growing bone is increasing in curvature, it is expected that the proliferative zone on the side of increasing convexity will be larger and more densely packed with chondrocytes than on the concave side.

Materials and Methods

Twenty-two Duroc boar × Landrace × Large White crossbred pigs which died in the normal course of farming operations were collected from a pig farm in Western Australia (UWA Animal Ethics Committee approval F 69199). The pigs ranged in age from eight days prenatal to 23 weeks postnatal. The ages of the postnatal pigs were obtained from the supplier whereas those of the prenatal pigs were derived by measuring crown-rump length (Ullrey et al., 1965). One additional 72-week-old pig was also used to compare the developing pigs with a mature adult. Each pig was weighed, the right radioulna removed and cleaned, and the bone length measured (Table 1). The radioulnae were photographed from the lateral side using a standardised orientation, ensuring the axis of the trochlea notch was vertical in each photograph. From the photographs, 16 landmarks and 80 semilandmarks were digitised in tpsDig (Rohlf, 2004). The landmarks were chosen to describe the overall shape of the bone, and the semilandmarks were used to characterise the curvature of the bones’ caudal and cranial surfaces in the sagittal plane (see Fig. 1; Table 2).

Table 1 The age and weight of the pigs and the lengths of the radioulnae used.

Age (weeks)	Weight (kg)	Length (mm)	
−1.14	0.90	50.1	
−0.57	1.06	63.4	
−0.43	0.95	57.1	
0	1.96	67.6	
0	1.14	55.4	
1	2.18	67.8	
2	2.66	69.1	
3	3.18	73.8	
5	8.10	100.8	
6	7.32	103.0	
7	11.30	104.3	
8	7.92	91.5	
9	19.98	121.5	
10	25.00	136.2	
12	25.00	134.6	
13	37.00	149.0	
17	57.00	168	
20	83.00	198	
22	100.00	201	
22	100.00	205	
22	98.00	185	
23	105.00	206	
72	180.00	285	

Figure 1 Diagram of a lateral view of the radioulna with the landmarks (numbered grey dots) and semilandmarks (black dots) used: 20× posterior ulna, 20× posterior radius and 40× anterior ulna.

Table 2 The landmarks on the radioulna.

No.	Definition	
1	Apex of olecranon	
2	Proximal epiphyseal plate at posterior margin of ulna	
3	Proximal epiphyseal plate at anterior margin of ulna	
4	Distal epiphyseal plate at posterior margin of ulna	
5	Distal epiphyseal plate at anterior margin of ulna	
6	Posterior-most point of distal surface of ulna	
7	Styloid process of ulna	
8	Styloid process of radius	
9	Distal epiphyseal plate at posterior margin of radius	
10	Middle point of anterior margin of radius	
11	Proximal epiphyseal plate at anterior margin of radius	
12	Proximal epiphyseal plate at posterior margin of radius (lateral)	
13	Midpoint of distal trochlear surface	
14	Lateral point of mid-trochlear surface	
15	Midpoint of mid-trochlear surface	
16	Midpoint of proximal trochlear surface	

The coordinates of the three sets of semilandmarks, describing the curvature of the posterior ulna and the anterior and posterior radial margins, were translated, scaled and rotated so the proximal semilandmark was at the origin (0,0) and the distal semilandmark had the coordinate (1,0) (after Richmond & Whalen, 2001). The largest Y-value among the semilandmarks represented the normalised curvature (i.e. maximum subtense/chord length).

The 96 landmarks were submitted to geometric morphometric analysis in morphologika (O’Higgins & Jones, 1998). First the landmark sets were Procrustes registered so differences in size, position and orientation among the landmarks were removed and only differences in the shape described by the landmarks remained. The principal components analysis was applied to identify the main patterns of shape variation. As we were interested in the changes in shape that accompany growth, form space was used so that the first principal component (PC1) characterised the shape change with increasing size.

Six specimens of different ages (one week prenatal and one, five, nine, 17 and 22 weeks postnatal) were selected and their epiphyseal growth plates examined. The bones were sectioned in the sagittal plane, and samples of the distal ulnar, and proximal and distal radial epiphyseal plates were removed for histological analysis. The proximal ulnar plate was not included because it is a traction epiphysis for the olecranon process and does not form part of the shaft of the ulna (Morgan, Wind & Davidson, 2000).

Each growth plate sample included the cranial and caudal sides of the epiphysis. The samples were fixed in 10% neutral-buffered formalin, rinsed in 0.9% saline and then immersed in rapid decalcifier (Apex Engineering Products Corp., Aurora, IL, USA) for between 2 and 4 h. When the samples were soft enough, they were rinsed in saline and agitated in phosphate-buffered saline (PBS) for 15 min, then placed in 30% sucrose solution made in PBS overnight. Samples were then agitated for 15 min in a 50:50 solution of PBS and OCT (optimal cutting temperature compound—ProSciTech, Thuringowa, QLD, Australia) before being embedded in OCT, frozen and sectioned at 40 μm using a cryostat at −20 °C (Leica CM 3050s; Leica, Wetzlar, Germany). Each block had 25 sections cut and every fifth section was used. The sections were placed on Superfrost Plus slides (Lomb Scientific Pty Ltd., Taren Point, NSW, Australia) and air dried overnight. Initially the slides were rinsed and stained in the normal vertical orientation, but this resulted in the loss of some sections (so only four of the six distal radial sections were available). To avoid section loss, subsequent slides were laid horizontally, drops of tap water were used to wash off the OCT, and Gill’s Haematoxylin I was applied for 1 min before rinsing and mounting with Kaiser’s aqueous mounting medium.

Images of the epiphyseal plates were captured at 100× magnification (Olympus BX50 microscope; Olympus Corporation, Tokyo, Japan). The images were exported to tpsDig, and the thickness of the proliferative zone was measured in five locations 100 μm apart, and the mean recorded (Fig. 2).

Figure 2 (A) An epiphyseal plate with the proliferative zone outlined in dashed lines.

Counting frames (green and red boxes) were placed equidistantly across the proliferative zone. The black lines represent the measured thickness of the proliferative zone. (B) Inset at higher magnification, showing which cell nuclei (yellow dots) in each counting frame were counted by the software (those wholly within the frame and those intersecting the green boundary).

Chondrocytes in the proliferative zone were counted using the optical fractionator probe in Stereo Investigator (MBF Bioscience, Williston, VT, USA). This probe is used to directly count proliferating chondrocytes within a defined volume, which then produces an unbiased population estimate without being affected by the size, distribution and orientation of the chondrocytes within this zone (West, Slomianka & Gundersen, 1991). The user identifies the region of interest (proliferative zone) in five sections of each tissue block and Stereo Investigator randomly selects 20 equidistant sampling sites in each section, so 100 sampling sites are counted for each tissue block (Fig. 2). From these estimates of chondrocyte population, cell density was estimated by dividing cell count by the total volume of the sampling sites/optical disectors (West, Slomianka & Gundersen, 1991).

Using the statistical package Genstat (18th edition, VSN International), Shapiro–Wilk tests were used to confirm normality of data. We then conducted linear regression to determine if the curvatures of the posterior ulna, posterior radius and anterior radius increased significantly with age. Paired t-tests were used to check for differences in the proliferative zone thickness and cell density between the cranial and caudal side of each epiphyseal growth plate.

Results

The age, weight and radioulnar length of the 23 pigs is shown in Table 1. These data are all highly correlated with one another (R > 0.95). The 72-week-old pig was excluded from the statistical analyses, but is shown in Fig. 3 to compare an adult pig with the developmental series.

Figure 3 PC1 and age.

Form space PC1 (98.5%) is plotted against age. Wireframe diagrams represent the shapes at the extremes of PC1 (−1 and +1), accompanied by representative photographs of neonatal and 23-week-old radioulnae (scale bars indicate 1 cm).

Figure 3 shows wireframe diagrams representing the extremes of PC1 in a form space analysis of radioulna shape based on the 96 landmarks and semilandmarks. PC1 accounted for 98.5% of the variation and showed a 0.97 correlation with age. This figure shows that the olecranon process becomes larger and the trochlear notch becomes relatively smaller with age. The overall curvature of the radioulna increases and this is particularly evident in the distal areas.

When the curvature measurements for the posterior ulna and the posterior and anterior margins of the radius are regressed against age, all three curves increase. This increase is only significant for the posterior ulna and anterior radial curves (Fig. 4). The curvature value for the 72-week-old pig is also indicated, and shows that the adult curvature is attained in these pigs by 23 weeks of age. Indeed, the anterior radial curve diminishes somewhat in the older pig.

Figure 4 Change in curvature with age.

Plots of the normalised curvature versus age for (A) posterior ulna, (B) posterior radius and (C) anterior radius. The values for the 72-week-old pig (grey circle) are included to show how curvatures change beyond 23 weeks, but this specimen was not included in the regression analyses. The P values show the significance of the regression analyses.

The mean thicknesses of the cranial and caudal sides of the epiphyseal proliferative zones are shown in Fig. 5. Paired t-tests show that the cranial side is significantly thicker than the caudal side in all three growth cartilages examined (P < 0.01). Figure 6 shows the chondrocyte density, and again paired t-tests show these cells are more densely packed on the cranial than the caudal side (P < 0.01). Photomicrographs of the cranial and caudal sides of the three cartilage growth plates are shown in Fig. 7. In each case, the cranial side is seen to have a greater chondrocyte density than the caudal side.

Figure 5 Thickness of the cranial and caudal regions of the proliferative zone in (A) distal ulnar, (B) proximal radial and (C) distal radial epiphyseal growth plates.

Individual lines represent a single specimen with age (in weeks) indicated. **Paired t-tests show that that the cranial is significantly thicker than the caudal side (P < 0.01).

Figure 6 Chondrocyte density in the cranial and caudal regions of the proliferative zone in (A) distal ulnar, (B) proximal radial and (C) distal radial epiphyseal growth plates.

Individual lines represent a single specimen with age (in weeks) indicated. **Paired t-tests show that that the cranial has a significantly denser chondrocyte population than the caudal side (P < 0.01).

Figure 7 Photomicrographs showing examples of the cranial and caudal growth cartilages from the distal ulna (A, B), proximal radius (C, D) and distal radius (E, F).

Sections (A–D) are 40 μm thick and stained with haematoxylin. Sections (E, F) are 8 μm thick and stained with haematoxylin and eosin. All sections are presented at the same magnification (see 100 μm scale bar).

Discussion

This study explored the ‘when’ and the ‘how’ of curvature development in the pig radioulna. Both the radius and ulna were found to be caudally curved in near-term foetuses, and this curvature was shown to increase in the first six months of life. Examination of the epiphyseal growth plates showed more chondrocyte activity in the cranial (convex) side than the caudal (concave) side. This establishes a mechanism for the formation and maintenance of bone curvature during development. The differential growth in the cranial and caudal sides of the epiphysis may occur in response to bending strains that place the cranial side in compression and the caudal side in tension (or less compression) (Frost, 1964; Milne, 2016).

The geometric morphometric analyses and regressions of curvature against age showed that radioulnar curvature increased between two weeks prenatal and 23 weeks postnatal. This was true for both the entire bone, as seen in the Geometric Morphometric (GM) analyses, and for the posterior ulnar and anterior radial margins individually. However, the posterior margin of the radius did not significantly increase in curvature over this period. The slope of the posterior radial curve regression line was shallower, and the variation among curve values greater, than in the other measured curvatures (see Fig. 4). It may be that the contact between the posterior ulna and the anterior radius impedes curvature development there, or that clear identification of the anterior surface of the radius was difficult and affected our measurements.

Here we examined pigs between two weeks prenatal and 23 weeks postnatal. The prenatal pigs already had some curvature in their radioulnae so it is still uncertain whether this bone is curved throughout foetal life, or is straight during earlier stages of gestation. The initial foetal curvature seen here may be caused by primitive muscle actions in utero, causing forelimb movements. Such movements are at their greatest frequency around 15 days before birth (Cohen et al., 2010). Examination of younger foetuses would determine whether the radioulna is indeed curved during early gestation, or if it begins as a straight bone prior to any muscle loading. It is also uncertain whether curvature changes any further beyond 23 weeks, but the 72-week-old specimen suggests that adult curvature is not greater than that of a 23-week-old. The geometric morphometric analysis (Fig. 2) and the linear regressions of curvature on age show that the 23-week-old pigs had already attained or even overshot the mature adult curvature.

Our analysis of the proliferative zone activity of the epiphyseal growth plates showed that this zone is thicker and more densely populated with chondrocytes on the cranial than the caudal side. In 1979, Frost proposed the chondral modelling theory—within the limits of physiological loading, cartilage grows faster under compression than tension, and grows slower under less compression than more. Numerous studies have supported this theory (e.g. Tardieu & Trinkaus, 1994; Urban, 1994; Hamrick, 1999; Congdon, Hammond & Ravosa, 2012; Rot et al., 2014). Asymmetrical growth of the epiphyseal plates can influence bone shape and was purported to be the mechanism behind both the normal formation of the carrying angle in human femora (Shackelford & Trinkaus, 2002) and the straightening of bones as fractures heal (Rot et al., 2014). Formation of bone curvature may also be attributable to such asymmetrical epiphyseal plate growth, as our results here suggest.

Recently, Henderson et al. (2017) showed that the ulnae of terrestrial primates and marsupials are caudally curved, like that of the pig and other terrestrial quadrupeds. Conversely, the ulna of arboreal primates and marsupials is cranially curved. Terrestrial quadrupeds’ forearm is dominated by the action of the triceps muscle acting to maintain extension of the elbow, but arboreal species rely on the action of the brachialis muscle to flex the elbow and allow the animal to climb and cling among the branches. The actions of triceps and brachialis place opposite bending strains upon the ulna, and it is proposed that the corresponding opposite curvatures are adaptations to those different bending strains (Milne, 2016). If bone and cartilage are stimulated to grow by compressive loading and growth is inhibited by tension (Frost, 1979), then this would provide a mechanism by which these opposite curvatures develop in terrestrial and arboreal species. In the case of the developing pig, where triceps acts to maintain stance during locomotion, the radioulna would experience cranial bending (with more compression on the cranial side). These bending strains may be responsible for the greater proliferation of cartilage cells in the cranial than the caudal side of the growth plates, and thus may be responsible for the development of the caudal curvature (caudal concavity) observed here.

Conclusion

Until recently, the timing and the mechanism of bone curvature formation was unknown. Lanyon (1980) saw a straightening of the rat tibia in the absence of muscle action but did not specify whether the bone curvature decreased or failed to form in the first place. The present study is the first to document that curvature does exist early in life, and increases as the animal grows.

Frost’s (1979) chondral modelling theory was used to explain how congruent joint surfaces are maintained and how the femoral carrying angle forms—however, until now this mechanism has not been explicitly associated with curvature formation. The results presented here suggest that chondral modelling is also applicable to the formation of bone curvature. Further, it seems likely that the formation of radioulnar curvature may be a direct result of the habitual action of the triceps muscle, inducing a strain gradient that drives chondral modelling in the epiphyseal growth plates of the growing bone. This was a pilot study exploring the development of curvature and a possible mechanism of curvature formation; it is hoped that future studies will test these ideas and further explore the mechanism of long bone curvature formation.

Supplemental Information

Supplemental Information 1 Raw data.

Click here for additional data file.

John Kopriwa provided the specimens used in the study. Lutz Slomianka, Mary Lee and Guy Ben-Ary assisted in the preparation and analysis of the cartilage samples. We thank Virginia Abdala and two anonymous reviewers for their comments and feedback, which improved this manuscript.

Additional Information and Declarations

Competing Interests

Author Contributions

Animal Ethics

Data Availability

The authors declare that they have no competing interests.

Jess Pantinople conceived and designed the experiments, performed the experiments, analysed the data, wrote the paper, prepared figures and/or tables, reviewed drafts of the paper.

Kyle McCabe analysed the data, wrote the paper, reviewed drafts of the paper.

Keith Henderson analysed the data, reviewed drafts of the paper.

Hazel L. Richards wrote the paper, prepared figures and/or tables, reviewed drafts of the paper.

Nick Milne conceived and designed the experiments, performed the experiments, analysed the data, contributed reagents/materials/analysis tools, wrote the paper, prepared figures and/or tables, reviewed drafts of the paper.

The following information was supplied relating to ethical approvals (i.e. approving body and any reference numbers):

The University of Western Australia Animal Ethics Committee (UWAAEC) provided approval for our use of already deceased animals (F69199).

The following information was supplied regarding data availability:

The raw data has been supplied as Supplemental Dataset Files.

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
