# Peer review of "The development of curvature in the porcine radioulna"

_PeerJ, doi:10.7717/peerj.3386_

## Round 0.1 · original submission · Major Revisions

I have now received two reviews of your manuscript. I agree with both reviewers, that the study is valuable but it requires more work. The reviewers provided several points that should be addressed in order to improve the ms. I am particularly concern by the ethical controls and permissions to perform your work, please provide them in the next round of the manuscript. There are several flaws in your methodological procedures that should be arranged, in particular you should provide good images and clear explanations of the methods for calculating chondrocytes density. Please note that your conclusions tend to go beyond your data, for example in relation to the triceps action. I really would appreciate a full consideration of all reviewer's suggestions.

Reviewer 1 ·

Basic reporting

Clear, professional English is used throughout. Some small areas for improvement are listed below.
Line 63: “healing chondrocytes” is not very scientific language
Lines 65, 212: clarify fracture healing of immature bones (rather than fracture healing in general)

Literature generally is well cited, but a few locations in which citations would strengthen statements are listed below.
Lines: 31, 64, 183- references needed

Some figures could be made easier to follow. I suggest that for Figures 3 to 5, ‘A-C’ is replaced by the names of the regions (e.g., Posterior Ulna).

Thank-you for supplying the raw data. However, the excel spreadsheet supplied needs to be significantly tidies up. It has many unlabelled graphs and appears to have data that appears to be of no relevance for the reader (e.g., Checklist)

The abstract could be made more clear, as suggested below:
Line 10: why is long bone curvature of interest?
Line 17: which three epiphyseal plates
Line 19: “Frost’s chondral modelling theory” would be more clear
Line 19: explain what the theory is (most people will not be aware)

Experimental design

Please provide details on the ethical controls and permissions for working on these specimens, including the fetal ones.

For paired t-tests, the differences should be approximately normally distributed. Were tests for normality performed? Please report on the normality of the differences.

I found it hard to follow the methodology when so many specific packages/methods (e.g., tpsDig, optional fractionator probe) that I am not familiar with are referred to. I suggest that the authors make the description of the methodology more accessible by explaining what the softwares do.

I also found it hard to follow some of the results, since key terms like “PC1” were not explained. What does the result given in line 151 mean in real terms (or plain English)?

In Figure 3, p-values are shown, but the text does not explain what these p-values are actually for.

The given age range for the fetal specimens does not match that of the table. How were the fetal ages known to such accuracy (days before birth)?

Why are there apparently gaps in the data- e.g., only 4 results in Fig 5 for the distal radial epiphysis? Such details should be provided in the methods/results.

Validity of the findings

My biggest concern with the findings is that no images of the chondrocyte quantification are provided, making it very difficult to judge the validity of the methodology or the results. At least one image per side per age group should be provided- as the authors have these images, why aren’t they shown?

Related to this, why were 20 sampling sites chosen- how were these chosen, why “approximately” 20, and why were cells in the whole area not counted?

My other main concern is with the authors’ conclusion that the action of the triceps leads to the curvature changes and activity of the epiphyseal chondrocytes. This is much too far of a conjecture that is not at all backed up by the study. All bones under muscular loading are subject to much more complex loading patterns than simply “longitudinal” or “oblique”. In order to propose that the action of a specific muscle is responsible for the results found, finite element modelling or similar methods would be required. I recommend that the authors avoid suggesting (in their abstract, discussion or conclusion) that loading from the triceps is responsible for their results, unless there is some other supporting evidence.

Reviewer 2 ·

Basic reporting

The authors are commended on their literature review and references for background/context. For readership unfamiliar with shape modeling/principal component analyses, a brief review of its application would be beneficial.

The way the data are presented is somewhat confusing and could be improved prior to publication. For example, an "overview" summary figure of the proposed model would help tie the story together. Additionally, the wireframe diagrams are difficult to interpret differences; colored or grayscale overlays could emphasize the differences in shape better.

The imaging of the chondrocytes and methods for calculating density of such is not well reported. Because these approaches are image-based, it is requested that histological images with annotation of cell counting analyses are included in the supplemental files.

Experimental design

The approach of 2-D analyses for shape modeling may be sufficient; the authors should reflect on how more technical, common approaches (e.g., 3-D micro-computed tomography) could improve their understanding/outcomes.

This experimental design is characteristic in nature and the hypotheses are not clearly defined. Low sample size limits statistical power, and the variability of the data are not well accounted for.

Method for imaging the radioulnae should be better described. Was there a standardized way of imaging these, or could slight rotations/curvature influence the imaging plane? Were definition of curvature locations validated based on slight modifications to the imaging plane?

Validity of the findings

"Chondral modelling theory would suggest that the cranial side of the bone is under more compression than the caudal side, and we conclude that this is due to the action of triceps extending the elbow by pulling on the olecranon process." The authors are encouraged to use caution with statements such as this, as they did not directly investigate the action of the triceps muscle on the olecranon. A simple study investigating unloading of the triceps, via targeted denervation, would provide this information.

Lack of limb development/embryonic data may limit the extent to which this data can be interpreted, as the caudal curvature is present close to birth. All samples used had curvature to begin with, so the initiation of curvature is not defined in this work.

Additional comments

I find this work to be of high importance; however, the approaches may be over simplified for the outcomes. Improved techniques that capitalize on the 3-dimensional nature of curvature in long bones is recommended; at least, an additional investigation into the frontal plane alignment could greatly bolster this work. Out of plane changes in curvature would not be captured in the single plane analysis, and more rigorous techniques (e.g., 3D microCT imaging) are suggested.

Additionally, major improvements to the way that the data are presented are suggested. The figures look rudimentary, and are not sufficient for describing the story.

---

## Round 0.2 · Minor Revisions

I am willing to accept your manuscript, but I still need one more clarification. Regarding this sentence: "The proximal ulnar plate was not included because it is a traction epiphysis for the olecranon process and does not form part of the shaft of the ulna" it seems to me that a quotation is needed here. I do not think that this is a fact but a hypothesis not totally accepted yet.

---

## Round 0.3 · accepted · Accept

Thank you for having taken into account the suggestions of the reviewers and mine